# Serum galectins as potential biomarkers of inflammatory bowel diseases

**Tony B. Yu**[1,2], **Susanna Dodd**[3], **Lu-Gang Yu**[2], **Sreedhar Subramanian**[2,4]*

**1** Edinburgh Medical School, University of Edinburgh, Edinburgh, Scotland, United Kingdom,
**2** Gastroenterology Research Unit, Institute of Translational Medicine, University of Liverpool, Liverpool, England, United Kingdom, **3** Department of Biostatistics, University of Liverpool, Liverpool, England, United Kingdom, **4** Department of Gastroenterology, Royal Liverpool University Hospital, Liverpool, England, United Kingdom

\* sreedhar.subramanian@rlbuht.nhs.uk

**Data Availability Statement:** All relevant data are within the manuscript and its Supporting Information files.

**Funding:** This study was sponsored by Royal Liverpool University Hospital (to SS) and a

## Abstract

The inflammatory bowel diseases (IBD), which include mainly Crohn's disease (CD) and ulcerative colitis (UC), are common chronic inflammatory conditions of the digestive system. The diagnosis of IBD relies on the use of a combination of factors including symptoms, endoscopy and levels of serum proteins such as C-reactive protein (CRP) or faecal calprotectin. Currently there is no single reliable biomarker to determine IBD. Galectins are a family of galactoside-binding proteins that are commonly altered in the circulation of disease conditions such as cancer and inflammation. This study investigated serum galectin levels as possible biomarkers in determining IBD and IBD disease activity. Levels of galectins-1, -2, -3, -4, -7 and -8 were analysed in 208 samples from ambulant IBD patients (97 CD, 71 UC) patients and 40 from healthy people. Disease activity was assessed using Harvey-Bradshaw Index for CD and simple clinical colitis activity index for UC. The relationship of each galectin in determining IBD and IBD disease activity were analysed and compared with current IBD biomarker CRP. It was found that serum level of galectin-1 and -3, but not galectins-2, -4, -7 and -8, were significantly higher in IBD patients than in healthy people. At cut-off of 4.1ng/ml, galectin-1 differentiated IBD from healthy controls with 71% sensitivity and 87% specificity. At cut-off of 38.5ng/ml, galectin-3 separated IBD from healthy controls with 53% sensitivity and 87% specificity. None of the galectins however were able to distinguish active disease from remission in UC or CD. Thus, levels of galectins-1 and -3 are significantly elevated in both UC and CD patients compared to healthy people. Although the increased galectin levels are not able to separate active and inactive UC and CD, they may have the potential to be developed as useful biomarkers for IBD diagnosis either alone or in combination with other biomarkers.

## Introduction

Inflammatory bowel diseases (IBD) are common chronic inflammatory conditions of the digestive system. IBD mainly include Crohn's disease (CD), which can affect any part of the

Wellcome Trust Vocation Scholarship (to TB Yu, Wellcome Trust 201871/Z/16/Z).

**Competing interests:** TBY, SD, NM and JR declare no conflicts of interest. SS has received speaker fee from MSD, Actavis, Abbvie, Dr Falk pharmaceuticals, Shire and received educational grant from MSD, Abbvie. LGY is an Academic Editor of PLOS ONE. This does not alter our adherence to PLOS ONE policies on sharing data and materials.

digestive tract, and ulcerative colitis (UC), which affects only the colon[1]. Both CD and UC typically cause symptoms of diarrhoea, abdominal pain, fatigue and weight loss[1]. The prevalence of IBD in Western countries is high and a recent Scottish study estimated a prevalence of 784/ 100, 000 population[2]. The diagnosis of IBD usually relies on a combination of symptoms, laboratory tests such as C-reactive protein (CRP), faecal calprotectin (FC) and endoscopic evaluation. Endoscopic evaluation is considered the gold standard and at present, there are no simple blood markers which accurately diagnose IBD[3].

IBD typically follow a relapsing-remitting pattern with periods of disease activity interspersed with quiescent disease[1]. IBD disease activity in clinical practice is often assessed in a composite manner by (i) evaluation of symptoms using validated indices such as the Crohn's disease activity index (CDAI) or the Harvey-Bradshaw index (HBI) for CD, and the simple clinical colitis activity index for UC; (ii) assessment of biomarkers such as C-reactive protein (CRP) or faecal calprotectin (FC) and (iii) endoscopic evaluation[3]. Endoscopic assessment is considered the gold standard for assessment of disease activity but repeated endoscopy is impractical, expensive and associated with poor patient acceptance and a potential for complications. CRP is currently the most used non-invasive serum IBD biomarker in clinic. However, CRP determination [4] is limited by poor specificity and sensitivity as (i) CRP level is altered in other acute inflammatory conditions in addition to CD and UC and (ii) up to 25% of patients, who have demonstrable disease activity determined by endoscopy have normal CRP values[5]. FC determination has far greater accuracy for detecting intestinal inflammation than CRP[6] but it requires collection of faecal sample which is cumbersome and has poor patient acceptability[7]. Thus, there is a need for more accurate and non-invasive biomarkers for establishing disease activity in IBD.

Galectins are a family of 15 (so far) mammalian galactoside-binding proteins that share a consensus amino acid sequence in their carbohydrate recognition domains (CRDs)[8]. Based on their structural differences, galectin family members are classified into three subgroups of proto-, chimeric- and tandem-repeat-types. The prototype galectins include galectins-1, 2, 5, 7, 10, 11, 13, 14 and 15, each containing a single CRD in the polypeptide sequences and often identified as non-covalently linked homodimer. The chimera-type galectin, which has only one member galectin-3, is composed of an unstructured non-lectin domain linked to a CRD and can form as multiple homotypic forms such as a pentamer. The tandem-repeat-type galectins include galectin-4, 6, 8, 9 and 12, each containing two different CRDs close to each end of the single polypeptide chain[9].

Galectins are widely expressed by human cells, in particular epithelial and immune cells. Several galectin members including galectins-1, -2, -3, -4 and -8 [10] are known to be expressed by the human intestinal epithelium[11, 12]. A large body of evidence have reported important roles of these galectins in colorectal cancer development, progression and metastasis (e.g. tumor transformation, cancer cell adhesion, invasion, migration and angiogenesis) by galectin interaction with a number of galactose-terminated glycans[13]. Serum levels of galectins such as galectins-2,-3, -4 and -8 are significantly higher in colon cancer patients than in healthy people [14] and promotes circulating tumour cell haematogenous dissemination in metastasis[14–17]. Interactions of galectins with galactose-terminated glycans on cell surface have also been implicated in the pathogenesis of IBD by induction of T cell apoptosis and NF-kappa B signaling [18–20], [19, 21]. Level of serum galectin-3 was reported to be higher in IBD patients compared to healthy controls[22]. However, it is not known whether levels of any other galectin members are also altered in the circulation of IBD patients and whether serum galectins could be used as potential biomarkers for determining IBD and disease activity.

In this study, we compared the serum levels of galectins-1, -2–3, -4, -7 and -8 in patients with UC, CD and healthy people and analyzed the relationship of galectin levels with disease activity in IBD patients.

## Materials and methods

### Materials

Recombinant human galectins-1, -2, -3, -4, -7 and -8, antibodies against human galectins-1, -2, -3, -4, -7 and -8 and biotinylated antibodies against human galectins-1, -2, -3, -4, -7 and -8 were obtained from R&D Systems (Abingdon, UK). ExtrAvidin peroxidase, FAST OPD and bovine serum albumin were from Sigma-Aldrich (Dorset, UK).

### Serum samples

Serum samples were obtained from a cross-sectional study of ambulant patients attending a specialist IBD clinic at the Royal Liverpool University Hospital. A total of 168 samples from IBD (including 97 from CD and 71 from UC) patients and 40 from healthy people were obtained. Ethical approval was obtained from the Liverpool Regional Ethics Committee. Written informed consent was obtained from all participants and the study population was comprised of adults over the age of 18. The diagnosis of IBD was established using conventional clinical, endoscopic and radiological criteria. All patients were eligible for inclusion excepting patients with an ostomy. Baseline details including disease phenotype, disease duration, smoking status, concurrent medications, prior surgical history and HBI for Crohn's disease and SCCAI for ulcerative colitis were collected on the day of enrolment. The HBI is a 5-item index used to assess disease activity in CD with a score $\leq$4 indicating remission, scores of 5–7, 8–16 and >16 indicating mild, moderate and severe disease[23]. The SCCAI includes 6 variables with a score of $\leq$2 indicating remission, scores of 3–5, 6–11 and $\geq$12 indicating mild, moderate and severe disease[24]. Laboratory tests were conducted as part of routine clinical care and included full blood count, renal and liver function tests, C-reactive protein (CRP) measurement. We investigated the ability of CRP to distinguish active disease from remission using a threshold of CRP$\geq$5mg/L, as described previously [25].

### Determination of serum levels of galectins

Serum levels of galectins-1, -2, -3, -4, -7 and -8, to which adequate capture and detection antibodies and recombinant galectins are available commercially for sandwich galectin ELISA, were determined by sandwich ELISA as described in our previous study[14]. Briefly, high-binding 96-well plates were coated with anti-galectin antibody at 2.5 μg/mL in coating buffer (15 mM $Na_2CO_3$ and 17 mM $NaHCO_3$, pH9.6) overnight at 4˚C. The plate was washed with washing buffer (0.05% Tween-20 in PBS) and incubated with blocking buffer (1% bovine serum albumin in PBS) for 1 hr at room temperature. Serum samples (1:2 or 1:5 dilution in PBS) or standard recombinant galectins were introduced to the plates for 2 hrs. After two wishes with PBS, biotinylated anti-galectin antibody (1.25 μg/mL in blocking buffer) was applied for 1 hr at room temperature before application of ExtrAvidin peroxidase (1:10,000 dilution in blocking buffer) for 1 hour. Following two washes with PBS, the plates were developed with SigmaFAST OPD for 10 minutes. The reaction was stopped by adding 4 mol/L sulfuric acid before the absorbance was read at 492 nm by a microplate reader.

## Statistical analysis

Differences between the levels of each galectin in UC, CD and healthy people were analysed by Mann-Whitney U test. Patient demographics are summarised as mean (SD) or median (IQR) for continuous variables and frequency (%) for categorical variables. Mean and median serum galectin levels were calculated to account for inter-individual variability in serum galectin levels. The difference between remission and active disease groups was analysed using Mann-Whitney test. Receiver operating characteristic (ROC) curves were constructed to assess the independent ability of galectins to discriminate between patients with and without clinical remission, using the area under the ROC curve (AUROC). The optimal cut-off point was calculated by minimising the distance between the ROC curve and the upper left hand corner of the plot. Analysis was conducted for all patients combined and then by diagnosis. Similar analyses were conducted to examine the correlation between CRP and disease activity. All analysis was performed using Stata v13 software (State Statistical Software: Release 12. College Station, TX: StataCorp LP), $p < 0.05$ is considered significant in all cases.

## Results

A total of 208 serum samples (40 from healthy people, 97 from CD and 71 from UC patients) were included in the study. The baseline characteristics of the included IBD patients and healthy people are summarised in Table 1. Of the IBD patients, 70 (72.2%) CD patients and 30 (43.3%) UC patients were in clinical remission as defined by HBI≤4 or SCCAI≤2.

## Serum galectin levels in healthy people and IBD patients

Serum levels of six galectin members were analysed. Among these galectin members, two galectins, galectin- 1 and -3 were found to have significantly higher levels in IBD patients compared to the healthy controls. When galectin levels were compared separately between UC and CD with healthy controls, significantly higher levels of galectin-1 (p = 0.0001 and 0.0003) and galectin 3 (p = 0.0003 and 0.0001, respectively) were found in UC and CD respectively (Table 2). Serum levels of galectin-2, -4, -7 and -8 showed no significant differences in UC and CD than in the healthy controls, although their levels in IBD patients were numerically higher. We next analysed the optimal concentration of galectins-1 and -3 that could distinguish IBD from healthy controls. At a cut-off of 4.1ng/ml, galectin-1 differentiated IBD from healthy controls with 71% sensitivity and 87% specificity. At a cut-off of 38.5ng/ml, galectin-3 differentiated IBD from healthy controls with 53% sensitivity and 87% specificity. Finally, we analysed serum levels of galectins-1, -2, -3, -4, -7, and -8 according to disease location (S1 File: S1-S6 Tables), disease duration (S1 File: S7-S12 Tables) and concomitant medications (S1 File: S13-S18 Tables). No significant differences in median galectin values across the variables was observed in any galectin member (S1 File: S1 to S18 Tables). As the sample size in each subgroup in this study was small, future studies with larger sample numbers would help to further clarify this conclusion.

## Serum galectins in active and inactive IBD

We next analysed the relationship of levels of galectin members in active and inactive CD and UC. None of the serum galectins showed significantly different in active UC and CD group compared to UC (Table 3) and CD (Table 4) groups in remission. When serum CRP levels were examined, significantly higher CRP was seen in patients with both active UC (P = 0.047, Table 3) and CD (P = 0.047, Table 4) compared to patients in remission. We also analysed serum galectin levels according to the clinical severity of disease activity (remission, mild,

**Table 1. Baseline characteristics of the IBD patients and control healthy people.**

|  |  | Controls (n = 40) | Crohn's disease (n = 97) | Ulcerative colitis (n = 71) |
|---|---|---|---|---|
| Age, mean (SD), years |  | 48.8 (15.6) | 40.9 (15.0) | 45.1 (18.7) |
| Sex | Male | 16 (40.0) | 36 (37.1) | 33 (46.5) |
|  | Female | 21 (52.5) | 61 (62.9) | 38 (53.5) |
| Smoking | Current | 5 (12.5) | 25 (25.8) | 12 (16.9) |
|  | Ex | 8 (20.0) | 18 (18.6) | 18 (25.4) |
|  | Never | 11 (27.5) | 54 (55.7) | 41 (57.8) |
|  | Missing | 16 (40.0) |  |  |
| Disease location |  |  | Colonic: 26 (26.8) | Proctitis: 14 (19.7) |
|  |  |  | Ileal: 34 (35.1) | Left-sided: 37 (52.1) |
|  |  |  | Ileocolonic: 27(27.8) | Pancolitis: 20 (28.2) |
|  |  |  | Perianal: 25 (25.8%) |  |
| Disease duration (years) <1 |  |  | 8 (8.3) | 7 (9.9) |
|  | 1 to 5 |  | 15 (15.5) | 22 (31.0) |
|  | 5 to 10 |  | 18 (18.6) | 40 (56.3) |
|  | >10 |  | 49 (50.5) | 0 (0) |
|  | Missing |  | 7 (7.2) | 2 (2.8) |
| Previous surgery |  |  | 35 (36.1) | 0 |
| Medications | 5-ASA |  | 26 (26.8) | 58 (81.7) |
|  | Thiopurines |  | 45 (46.4) | 19 (26.8) |
|  | Methotrexate |  | 2 (2.1) | 1 (1.4) |
|  | Anti-TNF agent |  | 19 (19.6) | 5 (7.0) |
|  | Corticosteroids |  | 6 (6.2) | 9 (12.7) |
| Disease activity | Active |  | 27 (27.8) | 41 (57.8) |
|  | Remission |  | 70 (72.2) | 30 (42.3) |
| C-reactive protein (mg/L) |  |  |  |  |
|  | >5 |  | 29 (29.9) | 21 (29.6) |
|  | ≤5 | 17 (100) | 51 (52.6) | 45 (63.4) |
|  | Missing |  | 17 (17.5) | 5 (7.0) |

Categorical variables: values are number (%)

moderate or severe). We did not identify an association between serum galectin levels and increasing disease severity (S1 File: S19-S24 Tables).

We then analysed if a combination of CRP with serum galectin-1 or -3 was better than either marker on its own in determining disease activity in UC and CD (Table 5). CRP in combination with either galectin-3 was able to distinguish active disease from remission in both UC (P = 0.0219) and CD (0.0056). However, the combination of galectin-3 with CRP was no better than CRP alone in distinguishing active disease from remission (p = 0.4150 from chi-square test comparing AUCs). In contrast to galectin-3, a combination of galectin-1 and CRP was unable to distinguish active disease from remission in both UC (P = 0.39) and UC ((P = 0.11). We then computed receiver operating characteristic (ROC) analysis of CRP alone or in a combination of galectins-1 and -3 and CRP in active and inactive UC and CD (Table 6). This combination did not show any significant improvement than CRP alone in distinguishing active from inactive disease.

**Table 2. Galectin levels in healthy controls, UC and CD patients.**

| | Controls (n = 40) | Crohn's disease (n = 97) | Ulcerative colitis (n = 71) | All IBD (n = 168) |
|---|---|---|---|---|
| **Gal-1** | | | | |
| N | 15 | 81 | 61 | 142 |
| Mean (SD), ng/ml | 6.2 (14.2) | 42.8 (194.8) | 51.7 (154.1) | 46.6 (177.9) |
| Median (range), ng/ml | 0.7 (0, 48.1) | 7.5 (0.3, 1300) | 11.9 (0, 815.9) | 9.2 (0,1300) |
| P value | | 0.0001 | 0.0002 | 0.0001 |
| **Gal-2** | | | | |
| N | 20 | 87 | 66 | 153 |
| Mean (SD), ng/ml | 177.7 (616.5) | 1068.2 (6229.7) | 2628.6 (12845.9) | 1741.3 (9650.1) |
| Median (range) | 0.6 (0, 2669.3) | 0.1 (0, 56371.4) | 0 (0, 99546.4) | 0.05 (0, 99546.4) |
| P value | | 0.6517 | 0.7962 | 0.7163 |
| **Gal-3** | | | | |
| N | 23 | 96 | 70 | 166 |
| Mean (SD), ng/ml | 22.0 (22.1) | 49.1 (38.8) | 55.3 (53.5) | 51.7 (45.5) |
| Median (range), ng/ml | 14.9 (0, 96.4) | 42.1 (0, 282.1) | 37.4 (0.7, 276.2) | 41.7 (0, 282.1) |
| P value | | 0.0003 | 0.0001 | <0.0001 |
| **Gal-4** | | | | |
| N | 23 | 88 | 71 | 159 |
| Mean (SD), ng/ml | 35.7 (58.0) | 45.2 (122.6) | 180.0 (989.7) | 105.4 (668.4) |
| Median (range),ng/ml | 0 (0, 185.5) | 0 (0, 868) | 0 (0, 8015) | 0 (0, 8015) |
| P value | | 0.9072 | 0.8028 | 0.8403 |
| **Gal-7** | | | | |
| N | 14 | 70 | 55 | 125 |
| Mean (SD), ng/ml | 132.2 (163.6) | 59.2 (138.7) | 52.4 (154.8) | 56.2 (145.4) |
| Median (range), ng/ml | 60.1 (0, 466.3) | 0 (0, 764.8) | 0 (0, 859.7) | 0 (0, 859.7) |
| P value | | 0.0350 | 0.1205 | 0.0574 |
| **Gal-8** | | | | |
| N | 18 | 93 | 69 | 162 |
| Mean (SD), ng/ml | 19.4 (31.0) | 48.3 (96.6) | 100.7 (313.5) | 70.6 (218.0) |
| Median (range), ng/ml | 4.7 (0, 104.2) | 3.5 (0, 578.5) | 18.1 (0, 2549.0) | 7.3 (0, 2549.0) |
| P value | | 0.103 | 0.1607 | 0.4197 |

## Discussion

In this study, we analysed serum levels of galectins-1, -2, -3, -4, -7 and -8 in IBD patients and healthy controls and investigated their value in determining IBD disease activity. It was found that among these six galectins, galectin-1 and -3 both showed significantly higher serum levels in UC and CD patients than in healthy people.

The intestinal epithelium harbours a number of galectin members and several of them are known to be altered in disease conditions in particularly cancer. Changes of the expression of galectins-1, -3, -4 and -9 in the inflamed tissues of IBD patients were reported previously[26]. Galectin profile analysis at mRNA level with biopsies from patients was able to distinguish IBD from other intestinal inflammatory disorders such as coeliac disease and potentially to also distinguish IBD from active form from quiescent ones[26]. Serum level of galectin-3 was reported in an early study to be higher in IBD patients irrespective of their disease activity[22] but no systematic comparison of the serum levels of galectin members has been done previously in IBD patients. The results in this study shows that levels of galectin-1 and -3, but not

**Table 3. Galectin levels in patients with active and inactive UC.**

| | UC only | | | P value |
|---|---|---|---|---|
| | SCCAI ≤2 | SCCAI >2 | All | |
| Galectin 1 | | | | |
| N | 25 | 36 | 61 | |
| Mean (SD), ng/ml | 104.9 (232.5) | 14.8 (14.3) | 51.7 (154.1) | |
| Median (range), ng/ml | 13.1 (0.2, 815.9) | 10.5 (0, 48.9) | 11.9 (0, 815.9) | 0.4545 |
| Galectin 2 | | | | |
| N | 25 | 41 | 66 | |
| Mean (SD), ng/ml | 2078.7 (6474.6) | 2964.0 (15578.7) | 2628.6 (12845.9) | |
| Median (range), ng/ml | 0.1 (0, 29589.9) | 0 (0, 99546.4) | 0 (0, 99546.4) | 0.3704 |
| Galectin 3 | | | | |
| N | 30 | 40 | 70 | |
| Mean (SD), ng/ml | 46.3 (43.1) | 62.0 (60.0) | 55.3 (53.5) | |
| Median (range), ng/ml | 28.3 (7.2, 186.4) | 47.1 (0.7, 276.2) | 37.4 (0.7, 276.2) | 0.2353 |
| Galectin 4 | | | | |
| N | 30 | 41 | 71 | |
| Mean (SD), ng/ml | 122.4 (457.6) | 222.1 (1248.2) | 180.0 (989.7) | |
| Median (range), ng/ml | 0 (0, 2494.7) | 0 (0, 8015) | 0 (0, 8015) | 0.4965 |
| Galectin 7 | | | | |
| N | 21 | 34 | 55 | |
| Mean (SD), ng/ml | 97.7 (226.6) | 24.4 (77.4) | 52.4 (154.8) | |
| Median (range), ng/ml | 0 (0, 859.7) | 0 (0, 439.4) | 0 (0, 859.7) | 0.1672 |
| Galectin 8 | | | | |
| N | 29 | 40 | 69 | |
| Mean (SD), ng/ml | 134.7 (468.5) | 76.1 (110.9) | 100.7 (313.5) | |
| Median (range), ng/ml | 19.7 (0, 2549.0) | 13.5 (0, 403.5) | 18.1 (0, 2549.0) | 0.6183 |
| CRP | | | | |
| N | 28 | 38 | 66 | |
| Mean (SD), mg/L | 4.8 (14.5) | 7.3 (12.5) | 6.3 (13.3) | |
| Median (range), mg/L | 0 (0, 56) | 0 (0, 55) | 0 (0, 56) | 0.0470 |

galectins-2, -4, -7 and -8, are significantly elevated in the circulation of IBD patients. The increased levels of galectin-1 or -3 however could not separate active and inactive forms of UC and CD. As an exploratory assessment for any indication of increased discrimination, a combination of CRP with galectins1 and 3 was further analysed but neither galectins-1 nor -3 were unable to provide additive sensitivity of CRP in distinguishing active disease from remission. Although further studies with larger sample size are needed to confirm this conclusion, it is possible that the increase in serum galectins might represent an early event in the inflammatory cascade and not subsequently rise further with increase of disease activity. The recent reports showing that several galectin members play key pathogenic roles in animal models of colitis[27] and galectin-3 caused macrophage activation in the induction phase of colitis in a dextran sodium sulphate animal model of colitis[28] are in line with this possibility.

The increased level of serum galectin-3 in IBD patients found in this study is in agreement with an early report showing higher serum galectin-3 level in IBD patients, irrespective of their disease activity, compared with healthy people[22]. It is noted that a recent study by Cibor et al reported significant elevation of serum level of galectin-3-binding protein (Gal-3BP), but not galectin-3 itself, in IBD patients[29]. The much lower median serum galectin-3 level in healthy people (7.1ng/ml) reported in the Cibor study than in this (22.0ng/ml) and a few other studies

**Table 4. Galectin levels in patients with active and inactive CD.**

| | CD only | | | P value |
|---|---|---|---|---|
| | HBI ≤5 | HBI >5 | All* | |
| Galectin 1 | | | | |
| N | 60 | 20 | 81 | |
| Mean (SD), ng/ml | 52.6 (225.7) | 15.1 (18.8) | 42.8 (194.8) | |
| Median (range), ng/ml | 7.5 (0.3, 1300) | 9.4 (0.6, 66.0) | 7.5 (0.3, 1300) | 0.5938 |
| Galectin 2 | | | | |
| N | 64 | 22 | 87 | |
| Mean (SD), ng/ml | 1336.6 (7226.3) | 336.0 (1198.4) | 1068.2 (6229.7) | |
| Median (range), ng/ml | 0.2 (0, 56371.4) | 0.1 (0, 5545.4) | 0.1 (0, 56371.4) | 0.6338 |
| Galectin 3 | | | | |
| N | 69 | 26 | 96 | |
| Mean (SD), ng/ml | 46.4 (41.5) | 54.0 (29.2) | 49.1 (38.8) | |
| Median (range), ng/ml | 40.1 (0, 282.1) | 53.4 (12.8, 118.1) | 42.1 (0, 282.1) | 0.1036 |
| Galectin 4 | | | | |
| N | 66 | 21 | 88 | |
| Mean (SD), ng/ml | 52.0 (133.0) | 25.8 (85.2) | 45.2 (122.6) | |
| Median (range), ng/ml | 0 (0, 868) | 0 (0, 390.5) | 0 (0, 868) | 0.3931 |
| Galectin 7 | | | | |
| N | 53 | 16 | 70 | |
| Mean (SD), ng/ml | 60.8 (148.5) | 57.8 (108.3) | 59.2 (138.7) | |
| Median (range), ng/ml | 0 (0, 764.8) | 0 (0, 419.6) | 0 (0, 764.8) | 0.8517 |
| Galectin 8 | | | | |
| N | 68 | 24 | 93 | |
| Mean (SD), ng/ml | 52.2 (102.4) | 39.2 (81.0) | 48.3 (96.6) | |
| Median (range), ng/ml | 4.0 (0, 578.5) | 0 (0, 287.1) | 3.5 (0, 578.5) | 0.2921 |
| CRP | | | | |
| N | 59 | 21 | 80 | |
| Mean (SD), mg/L | 4.6 (8.2) | 10.2 (12.1) | 6.1 (9.6) | |
| Median (range), mg/L | 0 (0, 35) | 6 (0, 38) | 0 (0, 38) | 0.0472 |

* one patient whose disease activity could not be accurately established was also included in this analysis.

[30] indicates that bigger cohorts of patients may be needed in future studies to clarify the discrepancy.

It is known that several galectin members including galectin-3[31] and galectin-1 [32, 33] can interact with cell surface glycans and induce endothelium, epithelium or T cell secretion of pro-inflammatory cytokines. For example, galectin-3 interaction with cell surface CD146 (MCAM) on vascular endothelial cells enhances endothelial secretion of IL-6, G-CSF and GM-CSF [34]. Interaction of galectin-1 with intestinal epithelial cells in the presence of TNFα, IL-13 or IL-5 was shown to enhance epithelial secretion of IL-10, IL-25, and TGF-β1[32]. The presence of higher concentrations of galectin-1 caused significant increase of Il-10 secretion from CD4(+) and CD8(+) T-cells[33]. As the chronic inflammatory condition in IBD is closely associated with cytokines[35], an increased secretion of those pro-inflammatory cytokines by circulating galectins-1 and -3, as a result of its elevated level in IBD, may play a role in disease initiation and progression. Moreover, an increased circulation of galectins may themselves have an impact on the pathogenesis and progression of IBD. Indeed, galectin-1 interaction with intestinal epithelial cells was altered in the presence of inflammatory stimuli and

**Table 5. Combination of galectins-1 and -3 and CRP in active and inactive UC and CD.**

|  | UC only | | | P value |
|---|---|---|---|---|
|  | SCCAI ≤2 | SCCAI >2 | All |  |
| Galectin 1+CRP |  |  |  |  |
| N | 23 | 33 | 56 |  |
| Mean (SD) | 119.5 (246.3) | 21.0 (21.1) | 61.4 (164.1) |  |
| Median (range), | 17.5 (0.2, 822.1) | 16.1 (0, 82.3) | 16.5 (0, 822.1) | 0.3910 |
| Galectin 3+CRP |  |  |  |  |
| N | 28 | 37 | 65 |  |
| Mean (SD) | 45.1 (44.0) | 63.1 (47.2) | 55.4 (46.3) |  |
| Median (range) | 28.3 (7.2, 219.8) | 58.0 (8.2, 233.7) | 36.2 (7.2, 233.7) | 0.0219 |
|  | CD only | | | P value |
|  | HBI ≤5 | HBI >5 | All |  |
| Galectin 1+CRP |  |  |  |  |
| N | 50 | 15 | 65 |  |
| Mean (SD) | 66.4 (246.6) | 30.4 (24.9) | 58.1 (216.6) |  |
| Median (range) | 14.6 (0.3, 1308) | 23 (0.6, 80.3) | 15.4 (0.3, 1308) | 0.1176 |
| Galectin 3+CRP |  |  |  |  |
| N | 59 | 21 | 80 |  |
| Mean (SD) | 46.3 (42.4) | 62.0 (28.1) | 50.4 (39.6) |  |
| Median (range) | 41.4 (0, 297.1) | 55.3 (20.6, 124.1) | 46.8 (0, 297.1) | 0.0056 |

modulated cell behaviours [32] and inflammation development[36]. Soluble galectin-3 produced by colon epithelial cells was reported to be a strong activator of colonic lamina propria fibroblasts by inducing cell NF-kappaB activation and IL8 secretion [12].

It should be mentioned that the disease activity used in this study was conventional clinical indices. Patients whose sera were analysed in this study did not undergo endoscopy or faecal collection. As clinical symptoms do not always correlate well with disease activity in IBD and as the clinical indices are less accurate than endoscopy and faecal calprotectin analysis, further investigations to include endoscopy and faecal calprotectin analysis will be needed to determine the possibility of using serum galectins as potential biomarkers in determining IBD disease activity. Further investigations can also include patients with gastrointestinal symptoms

**Table 6. ROC analysis of CRP alone or a combination of galectin 3 and CRP in active and inactive disease.**

|  | UC only | CD only | UC + CD |
|---|---|---|---|
| **CRP** | 0.645 (0.540, 0.749) [N = 65] | 0.642 (0.519, 0.765) [N = 80] | 0.621 (0.542, 0.700) [N = 145] |
| **Galectin 3+CRP** | 0.667 (0.529, 0.805) [N = 65] | 0.705 (0.579, 0.830) [N = 80] | 0.668 (0.579, 0.756) [N = 145] |
| **Galectin 3+CRP (< = />52)** | 0.637 (0.517, 0.756) [N65] | 0.698 (0.580, 0.816) [80] | 0.667 (0.588, 0.746) [145] |
| **Galectin 1+CRP** | 0.432 (0.275, 0.589) [56] | 0.634 (0.457, 0.811) [65] | 0.517 (0.408, 0.625) [121] |
| **Galectin 1+CRP (</> = 17.9)** | 0.488 (0.353, 0.624) [56] | 0.623 (0.482, 0.765) [65] | 0.541 (0.450, 0.633) [121] |

Cut-off with highest correct % classification is 17.9 and 52 for galectin-1 and -3 respectively (for all IBD patients)–hence values above relate to ROC curve for 'CRP + galectin1 > = 17.9' versus 'CRP > 5' and CRP + galectin3 > = 52' versus 'CRP > 5'. The numerical value for the combination of serum CRP and galectin was obtained by simply combining the actual values.

such as irritable bowel syndrome and coeliac disease to truly establish the discriminatory value of galectins. Due to the limited number of patients in each sub-group, meaningful evaluation of the impact of disease location (e.g. ileal compared to colonic Crohn's disease), disease duration and medications on serum galectin values was not possible in this study.

In conclusion, serum levels of galectins-1 and -3, but not galectins-2, -4, -7 and -8, are significantly elevated in both UC and CD patients in comparison to healthy people. Although the increased levels of these galectins are not able to separate active and inactive UC and CD, they have the potential to be developed as biomarkers for general IBD determination.

## Supporting information

**S1 File. Galectin-1, -2, -3, -4, -7, -8 expression according to disease duration (S1-S6 Tables), disease location (S7-S12 Tables), medications (S13-18 Tables) and disease severity (S19-S24 Tables).**
(DOCX)

## Acknowledgments

This study was supported by a Wellcome Trust Biomedical Vacation Scholarship (to TB Yu) and Royal Liverpool University Hospital.

## Author Contributions

**Conceptualization:** Lu-Gang Yu, Sreedhar Subramanian.

**Data curation:** Tony B. Yu.

**Formal analysis:** Tony B. Yu, Susanna Dodd, Sreedhar Subramanian.

**Funding acquisition:** Lu-Gang Yu, Sreedhar Subramanian.

**Investigation:** Tony B. Yu, Susanna Dodd, Lu-Gang Yu, Sreedhar Subramanian.

**Methodology:** Tony B. Yu, Susanna Dodd, Lu-Gang Yu.

**Project administration:** Lu-Gang Yu.

**Supervision:** Lu-Gang Yu, Sreedhar Subramanian.

**Writing – original draft:** Lu-Gang Yu, Sreedhar Subramanian.

**Writing – review & editing:** Lu-Gang Yu, Sreedhar Subramanian.

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
