## [Decision Letter · Decision Letter 0]

20 Sep 2019

PONE-D-19-23084

SERUM GALECTINS AS POTENTIAL BIOMARKERS OF INFLAMMATORY BOWEL DISEASES

PLOS ONE

Dear Dr Subramanian,

Thank you for submitting your manuscript to PLOS ONE. After careful consideration, we feel that it has merit but does not fully meet PLOS ONE’s publication criteria as it currently stands. Therefore, we invite you to submit a revised version of the manuscript that addresses the very constructive points raised during the review process by the two reviewers.

We would appreciate receiving your revised manuscript by Nov 04 2019 11:59PM. To enhance the reproducibility of your results, we recommend that if applicable you deposit your laboratory protocols in protocols.io, where a protocol can be assigned its own identifier (DOI) such that it can be cited independently in the future. For instructions see: http://journals.plos.org/plosone/s/submission-guidelines#loc-laboratory-protocols

We look forward to receiving your revised manuscript.

Kind regards,

Roger Chammas, M.D, Ph.D

Academic Editor

PLOS ONE

2. Please provide additional details regarding participant consent. In the ethics statement in the Methods and online submission information, please ensure that you have specified (1) whether consent was informed and (2) what type you obtained (for instance, written or verbal). If your study included minors, state whether you obtained consent from parents or guardians. If the need for consent was waived by the ethics committee, please include this information.

"TBY, SD, NM and JR declare no conflicts of interest.

SS has received speaker fee from MSD, Actavis, Abbvie, Dr Falk pharmaceuticals, Shire and received educational grant from MSD, Abbvie.

LGY is an Academic Editor of PLOS ONE"

Reviewers' comments:

Reviewer's Responses to Questions

**Comments to the Author**

1. Is the manuscript technically sound, and do the data support the conclusions?

Reviewer #1: Partly

Reviewer #2: Partly

2. Has the statistical analysis been performed appropriately and rigorously? 

Reviewer #1: Yes

Reviewer #2: Yes

3. Have the authors made all data underlying the findings in their manuscript fully available?

Reviewer #1: Yes

Reviewer #2: Yes

4. Is the manuscript presented in an intelligible fashion and written in standard English?

Reviewer #1: Yes

Reviewer #2: Yes

5. Review Comments to the Author

Reviewer #1: The authors in this study determine the potential for serum galectin-1 and galectin-3 to act as a biomarker for either IBD screening or IBD activity. Previously, tissue levels of galectin-1 and galectin 3 have been shown to be elevated in IBD patients but a systematic comparison of galectin-1 and galectin-3 with other biomarkers has not occurred.

I am concerned by the analysis that uses a CRP threshold of 5mg/dL. The reference that they cite to provide this actually provides data to support a threshold of 5mg/L. This does not negate the findings that serum levels of galectin-1 and -3 are increased in IBD patients but it does affect much of the data analysis.

They do not really describe the issue with these markers being elevated in IBD but not acting as markers of disease activity. Discussion should center around the possibilities that either it is only an early step in the inflammatory cascade or that treatment effect also could cause this to falsely rise (rather than be a marker of inflammation).

Minor points include a few spelling errors and that much of the data provided in the introduction is not appropriately referenced.

Reviewer #2: GENERAL COMMENTS:

The authors presented a relevant work for the field of inflammatory bowel diseases (IBD), such as Crohn's disease and ulcerative colitis. They studied whether circulating galectins can be associated with C-reactive protein in diagnosis and/or biomarkers of IBD. In this study, it was evident that serum galectin-1 and -3 levels were increased in serum of patients with IBD in comparison with healthy individuals. On the other hand, the levels of galectin-1, -2, -3, -4, -7 and -8 did not modified in these IBD patients. However, the high levels of galectins were not able of separate active and inactive IBD. The potential to use galectin levels as biomarkers was clearly demonstrated in this manuscript, but the authors need to improve the analysis to try to validate their proposal. Current format is not recommended to publish, although the authors should be encouraged to resubmit after major revisions.

SPECIFIC POINTS PER SECTIONS:

INTRODUCTION

1. The first paragraph needs some references.

2. Some references about galectins and IBD were neglected in the manuscript, such as Sundblad et al., 2018 and Cibor et al., 2019. Please, explore this theme in the introduction section. Moreover, Cibor and colleagues concluded that serum galectin-3 (and Gal-9) should not be considered biomarkers of IBD. Please, make a discussion regarding to galectins as biomarkers of IBD using more recent papers.

MATERIALS AND METHODS

1. Subtitle of the first paragraph is absent

RESULTS

1. Table 1 was appropriately used as base to correlate galectins levels with disease activity and C-reactive protein. However, to amplify the spectrum of analyzed data, the authors should explore underestimated data, such as disease location, disease duration, and medications:

1.1 - Galectin expression and synthesis vary throughout the small and large intestines. Then, disease location can be a relevant factor to be included in the analysis.

1.2 – Disease duration can also be important to study of biomarkers. Galectins have differential expression depending on stage of diseases, such as inflammation and fibrosis.

1.3 – Medications can alter the inflammatory and immunological responses and consequently the expression, synthesis and/or secretion of galectins. Medications can be critical modulators of biomarkers and it is impossible to be neglected in study of biomarkers.

1.4 - Values of serum levels of galectins are missed without a concentration unit as a reference (e.g. ng/mL).

2. Table 2 was excellent to detect serum levels of galectins, in special galectin-1 and -3. However, some information is missing:

2.1 - The insertion of concentration unit (possibly ng/mL) can improve to readers.

2.2 – In table legend, please, include the meaning of “Gal-1 conc”, “Gal-2 conc”, etc.

3. Table 3 was critical to compare active and inactive UC. Again, the insertion of concentration unit (possibly ng/mL) can improve to readers.

4. Table 4 was important to compare active and inactive CD. To better comprehension:

4.1 – Insert the concentration unit (possibly ng/mL) in values of serum galectins;

4.2 – The third column (“All”) shows an asterisk (*). Please, describe the meaning of this symbol.

5. Table 5 showed a comparison with different protocols of analysis using Gal-1 and Gal-3 independently combined with CRP.

5.1 – Describe the unit (possibly ng/mL);

5.2 – It was not clear the strategy to obtain the values (“mean” and “median”). Please, describe in the text or in materials and methods section

6. Table 6 showed a ROC analysis. However, the text subsequent (below of table) can be reorganized to explain the values (unit) and symbols (“>=” is equivalent to “≥”?).

DISCUSSION

1. Selection criteria for using galectin-1, -2, -3, -4, -7 and -8 should be discussed here. Why other galectins were not investigated in IBD patients? At least, these arguments can be detailed in materials and methods section.

2. The authors discussed about immunomodulatory roles of galectin-3 inducing pro-inflammatory signals (such as IL-6 dependent manner). These aspects can be explored with more details, including galectin-1 in the context. Several cytokines are associated with galectin-1 and galectin-3 levels in IBD. Imbalances between pro and anti-inflammatory responses seem relevant to insert both galectins as biomarkers.

3. There are several neglected works using galectins to study the pathogenesis and progression of IBD. The insertion in discussion section will improve the quality of this manuscript.

6. PLOS authors have the option to publish the peer review history of their article (what does this mean?). If published, this will include your full peer review and any attached files.

Reviewer #1: No

Reviewer #2: No

---

## [Author Response · Author response to Decision Letter 0]

22 Oct 2019

Dear Dr. Chammas,

Thanks for sending us the reviewers’ very helpful comments. We have made substantial revision of the manuscript (including the addition of 18 new tables as supplementary supporting materials and also addition of 11 new references) according to the referees’ suggestions and comments including the addition of 18 new tables in supplement materials. Taking the reviewers’ points in turns:

We have carefully edited the manuscript in accordance to the style requirements of PLOS One

2. Please provide additional details regarding participant consent. In the ethics statement in the Methods and online submission information, please ensure that you have specified (1) whether consent was informed and (2) what type you obtained (for instance, written or verbal). If your study included minors, state whether you obtained consent from parents or guardians. If the need for consent was waived by the ethics committee, please include this information.

Written informed consent was obtained from all participants in this study and the study population was comprised of adults over the age of 18. We have included these statements in the Method section (page 7).

"TBY, SD, NM and JR declare no conflicts of interest.

SS has received speaker fee from MSD, Actavis, Abbvie, Dr Falk pharmaceuticals, Shire and received educational grant from MSD, Abbvie.

LGY is an Academic Editor of PLOS ONE"

This has been revised according to the suggestions (p19)

Updated Competing Interest statement is included: TBY, SD, NM and JR declare no conflicts of interest. SS has received speaker fee from MSD, Actavis, Abbvie, Dr Falk pharmaceuticals, Shire and received educational grant from MSD, Abbvie. LGY is an Academic Editor of PLOS ONE. This does not alter our adherence to PLOS ONE policies on sharing data and materials.

Reviewer 1 comments:

I am concerned by the analysis that uses a CRP threshold of 5mg/dL. The reference that they cite to provide this actually provides data to support a threshold of 5mg/L. This does not negate the findings that serum levels of galectin-1 and -3 are increased in IBD patients but it does affect much of the data analysis.

They do not really describe the issue with these markers being elevated in IBD but not acting as markers of disease activity. Discussion should center around the possibilities that either it is only an early step in the inflammatory cascade or that treatment effect also could cause this to falsely rise (rather than be a marker of inflammation).

Thanks the reviewer for spotting the Unit typo which has been corrected in the revision (5mg/L)

As suggested, we have added the below paragraph to discuss the discovery that galectins are elevated in IBD but are not effective markers of disease activity. “Although further studies with larger sample size are needed to confirm this conclusion, it is possible that the increase in serum galectins might represent an early event in the inflammatory cascade and not subsequently rise further with increase of disease activity. The recent reports showing that several galectin members play key pathogenic roles in animal models of colitis [new ref 27] and galectin-3 caused macrophage activation in the induction phase of colitis in a dextran sodium sulphate animal model of colitis[new ref 28] are in line with this possibility” (p17).

Minor points include a few spelling errors and that much of the data provided in the introduction is not appropriately referenced.

All the spelling errors in the manuscript have been corrected and three new references (refs 1 to 3) have been added in the introduction (p4) 

Reviewer #2: GENERAL COMMENTS:

INTRODUCTION

1. The first paragraph needs some references. 

Several new references were added (refs 1-4, p4)

2. Some references about galectins and IBD were neglected in the manuscript, such as Sundblad et al., 2018 and Cibor et al., 2019. Please, explore this theme in the introduction section. Moreover, Cibor and colleagues concluded that serum galectin-3 (and Gal-9) should not be considered biomarkers of IBD. Please, make a discussion regarding to galectins as biomarkers of IBD using more recent papers.

These suggested papers have all been discussed in the study and references included 

MATERIALS AND METHODS

1. Subtitle of the first paragraph is absent

A subtitle of “Materials” has been added in the revision (p7)

RESULTS

1. Table 1 was appropriately used as base to correlate galectins levels with disease activity and C-reactive protein. However, to amplify the spectrum of analyzed data, the authors should explore underestimated data, such as disease location, disease duration, and medications:

We have conducted substantial additional analysis of serum galectin levels in accordance to disease location, disease duration and medications and included the results as 18 supplemental tables (Tables S1 to S18). These analyses showed no significant differences in median serum galectin values of any of the galectin members- 1, -2, -3, -4, -7 and -8 when stratified by disease location, duration or medications. We have included the results of those new analysis as supplement materials (Table S1 to S18) and add a statement in the main text as “We also analysed the serum levels of galectins-1, -2, -3, -4, -7, and -8 according to disease location (Table S1-S6), disease duration (Table S7-S12) and concomitant medications (Table S13-S18). No significant differences in median galectin values across the variables was observed for any galectin members (Table S1 to S18). As the sample size in each sub-group in this study was small, future studies with larger sample numbers would help to clarify this conclusion” (p11)

1.1 - Galectin expression and synthesis vary throughout the small and large intestines. Then, disease location can be a relevant factor to be included in the analysis.

1.2 – Disease duration can also be important to study of biomarkers. Galectins have differential expression depending on stage of diseases, such as inflammation and fibrosis.

1.3 – Medications can alter the inflammatory and immunological responses and consequently the expression, synthesis and/or secretion of galectins. Medications can be critical modulators of biomarkers and it is impossible to be neglected in study of biomarkers.

All have been revised as discussed above

1.4 - Values of serum levels of galectins are missed without a concentration unit as a reference (e.g. ng/mL).

Values of the serum galectin levels was added (ng/ml, Tables 2-4)

2. Table 2 was excellent to detect serum levels of galectins, in special galectin-1 and -3. However, some information is missing:

2.1 - The insertion of concentration unit (possibly ng/mL) can improve to readers.

2.2 – In table legend, please, include the meaning of “Gal-1 conc”, “Gal-2 conc”, etc.

3. Table 3 was critical to compare active and inactive UC. Again, the insertion of concentration unit (possibly ng/mL) can improve to readers.

Galectin concentration unit (ng/ml) was added and the word “conc” was removed to make it clearer 

4. Table 4 was important to compare active and inactive CD. To better comprehension:

4.1 – Insert the concentration unit (possibly ng/mL) in values of serum galectins;

4.2 – The third column (“All”) shows an asterisk (*). Please, describe the meaning of this symbol.

Concentration unit (ng/ml) was included and a footnote was added for the asterisk “One patient whose disease activity could not be accurately established was also included in this analysis” 

5. Table 5 showed a comparison with different protocols of analysis using Gal-1 and Gal-3 independently combined with CRP.

5.1 – Describe the unit (possibly ng/mL);

Concentration unit (ng/ml) was added

5.2 – It was not clear the strategy to obtain the values (“mean” and “median”). Please, describe in the text or in materials and methods section

This has now been clarified in the methods section with the addition of the following sentence: Mean and median serum galectin levels were calculated to account for inter-individual variability in serum galectin levels (p8). 

5. Table 6 showed a ROC analysis. However, the text subsequent (below of table) can be reorganized to explain the values (unit) and symbols (“>=” is equivalent to “≥”?).

We have now clarified this by the addition of the following to the footnote of the table: The numerical value for the combination of serum CRP and galectin was obtained by simply combining the actual values.

DISCUSSION

1. Selection criteria for using galectin-1, -2, -3, -4, -7 and -8 should be discussed here. Why other galectins were not investigated in IBD patients? At least, these arguments can be detailed in materials and methods section.

The selection of these galectin members was entirely due to the commercial availability of the detection agents (capture and detection anti-galectin antibodies and recombinant galectins) of the sandwich galectin ELISA. We have made this clearer in the Method section (p8)

2. The authors discussed about immunomodulatory roles of galectin-3 inducing pro-inflammatory signals (such as IL-6 dependent manner). These aspects can be explored with more details, including galectin-1 in the context. Several cytokines are associated with galectin-1 and galectin-3 levels in IBD. Imbalances between pro and anti-inflammatory responses seem relevant to insert both galectins as biomarkers.

As suggested, we have added three new paragraphs in the Discussion section to discuss the immunomodulatory roles of galectin-1 and -3 and the galectin-associated cytokines in IBD (p17-19) and included a few new references (refs 27-30).

3. There are several neglected works using galectins to study the pathogenesis and progression of IBD. The insertion in discussion section will improve the quality of this manuscript.

As suggested, we have discussed the few early studies about the role of galectin-1 and -3 in IBD pathogenesis and progression in the discussion section and included a few new references (ref 27 to 31)

We hope the addition of the new data and the substantial revision has satisfactorily addressed the reviewers’ comments and the manuscript is now in an acceptable form for publication by PLOS One.

Yours sincerely, 

Sreedhar Subramanian

---

## [Decision Letter · Decision Letter 1]

5 Nov 2019

PONE-D-19-23084R1

SERUM GALECTINS AS POTENTIAL BIOMARKERS OF INFLAMMATORY BOWEL DISEASES

PLOS ONE

Dear Dr Subramanian,

Thank you for submitting your manuscript to PLOS ONE. After careful consideration, we feel that it has merit but does not fully meet PLOS ONE’s publication criteria as it currently stands. Therefore, we invite you to submit a revised version of the manuscript that addresses the points raised during the review process.

We would appreciate receiving your revised manuscript by Dec 20 2019 11:59PM. To enhance the reproducibility of your results, we recommend that if applicable you deposit your laboratory protocols in protocols.io, where a protocol can be assigned its own identifier (DOI) such that it can be cited independently in the future. For instructions see: http://journals.plos.org/plosone/s/submission-guidelines#loc-laboratory-protocols

We look forward to receiving your revised manuscript.

Kind regards,

Roger Chammas, M.D, Ph.D

Academic Editor

PLOS ONE

Additional Editor Comments (if provided):

Reviewer 1 still has some concerns that need to be addressed before the ms is considered acceptable. Please, address the constructive issues raised and submit a revised version of your ms.

Reviewers' comments:

Reviewer's Responses to Questions

**Comments to the Author**

1. If the authors have adequately addressed your comments raised in a previous round of review and you feel that this manuscript is now acceptable for publication, you may indicate that here to bypass the “Comments to the Author” section, enter your conflict of interest statement in the “Confidential to Editor” section, and submit your "Accept" recommendation.

Reviewer #1: (No Response)

Reviewer #2: All comments have been addressed

2. Is the manuscript technically sound, and do the data support the conclusions?

Reviewer #1: Partly

Reviewer #2: Yes

3. Has the statistical analysis been performed appropriately and rigorously? 

Reviewer #1: Yes

Reviewer #2: Yes

4. Have the authors made all data underlying the findings in their manuscript fully available?

Reviewer #1: Yes

Reviewer #2: Yes

5. Is the manuscript presented in an intelligible fashion and written in standard English?

Reviewer #1: Yes

Reviewer #2: Yes

6. Review Comments to the Author

Reviewer #1: In the materials and methods section, there is a statement that "CRP>=5mg/L was used to define active disease" although I believe the clinical indices were actually used to define activity or inactivity.

I'm a bit concerned about the methods to add CRP and galectin-1 or galectin-3 to create a composite score to analyze against active or inactive IBD as it might underestimate the potential finding regarding an improvement in sensitivity for active IBD if you simply use the dichotomous variable of "galectin-1 normal or abnormal" and similarly for galectin-3.

With regard to the interesting finding that neither galectin-1 or galectin-3 associate with disease activity, it's not clear what the spectrum of disease activity (based on HBI and SCCAI) was. If there is a predominance of mildly active patients, that may explain this finding as patients with moderate-severe activity might be more likely to have higher serum biomarkers than patients in the inactive state.

Reviewer #2: The authors followed all suggestions. Current form is extremely relevant to review possible correlation between galectins and inflammatory bowel diseases (IBD), such as Crohn's disease and ulcerative colitis. This version contemplates an extensive analysis of data regarding to circulating galectins and biomarkers of IBD. Discussion about increase of galectin-1 and -3 serum levels in patients with IBD was substantially improved in comparison with first version of the review. The potential to use galectin levels as biomarkers was clearly demonstrated in this manuscript. Given that authors satisfactorily responded the reviewer's comments, added the relevance of the theme, current format can be recommended to publication.

7. PLOS authors have the option to publish the peer review history of their article (what does this mean?). If published, this will include your full peer review and any attached files.

Reviewer #1: No

Reviewer #2: No

---

## [Author Response · Author response to Decision Letter 1]

26 Nov 2019

Dear Dr. Chammas,

Thanks for giving us the opportunity to address the reviewers’ further comments to this manuscript. We have revised the manuscript in accordance to these comments and included six additional tables in the supplement materials. Take these comments in turns:

Reviewer 2:

In the materials and methods section, there is a statement that "CRP>=5mg/L was used to define active disease" although I believe the clinical indices were actually used to define activity or inactivity.

Clinical indices were indeed used to define disease activity. We however also used a CRP threshold of >=5mg/L to compare the diagnostic sensitivity of CRP compared to serum galectins in differentiation of active from inactive disease. To make this clearer, we have added below in the revision: ‘We investigated the ability of CRP to distinguish active disease from remission using a threshold of CRP≥5mg/L, as described previously’ in the method section (page 7). (please also see responses below).

I'm a bit concerned about the methods to add CRP and galectin-1 or galectin-3 to create a composite score to analyze against active or inactive IBD as it might underestimate the potential finding regarding an improvement in sensitivity for active IBD if you simply use the dichotomous variable of "galectin-1 normal or abnormal" and similarly for galectin-3.

We acknowledge the limitations of combining serum CRP to galectins-1 and -3 to produce a composite measure hence included the below in the discussion section to make it clearer the exploratory nature of this combination of analysis: “As an exploratory assessment for any indication of increased discrimination, a combination of CRP with galectins1 and 3 was further analysed but neither galectins-1 nor -3 were unable to provide additive sensitivity of CRP in distinguishing active disease from remission” (p17).

With regard to the interesting finding that neither galectin-1 or galectin-3 associate with disease activity, it's not clear what the spectrum of disease activity (based on HBI and SCCAI) was. If there is a predominance of mildly active patients, that may explain this finding as patients with moderate-severe activity might be more likely to have higher serum biomarkers than patients in the inactive state.

To address this comment, we have now performed additional analyses of serum galectins-1 and -3 across the spectrum of disease activity. We used previously defined thresholds of Harvey-Bradshaw index and simple clinical colitis activity index to define clinical severity. Both in CD and UC, there were approximately equal number of patients with mild and moderate disease activity respectively. Only one patient had severe disease. We have included the results of these analysis in six new tables in the supplemental materials (Tables S19-S24) and added below in in the results section: ‘We also analysed serum galectin levels according to the clinical severity of disease activity (remission, mild, moderate or severe). No association was observed between serum galectin levels and increasing disease severity” (p12). 

We hope these revisions have satisfactorily addressed all the reviewers’ comments and the manuscript is now in an acceptable form for publication. 

With many thanks,

Sreedhar Subramanian

---

## [Decision Letter · Decision Letter 2]

9 Dec 2019

PONE-D-19-23084R2

SERUM GALECTINS AS POTENTIAL BIOMARKERS OF INFLAMMATORY BOWEL DISEASES

PLOS ONE

Dear Dr Subramanian,

Thank you for submitting your manuscript to PLOS ONE. After careful consideration, we feel that it has merit but does not fully meet PLOS ONE’s publication criteria as it currently stands. Therefore, we invite you to submit a revised version of the manuscript that addresses the points raised during the review process. Specifically, consider the points raised by the reviewer 1 which points to some limitations and concerns about the score suggested and the relation amon CRP and galectins -1 and -3.

We would appreciate receiving your revised manuscript by Jan 23 2020 11:59PM. To enhance the reproducibility of your results, we recommend that if applicable you deposit your laboratory protocols in protocols.io, where a protocol can be assigned its own identifier (DOI) such that it can be cited independently in the future. For instructions see: http://journals.plos.org/plosone/s/submission-guidelines#loc-laboratory-protocols

We look forward to receiving your revised manuscript.

Kind regards,

Roger Chammas, M.D, Ph.D

Academic Editor

PLOS ONE

Reviewers' comments:

Reviewer's Responses to Questions

**Comments to the Author**

1. If the authors have adequately addressed your comments raised in a previous round of review and you feel that this manuscript is now acceptable for publication, you may indicate that here to bypass the “Comments to the Author” section, enter your conflict of interest statement in the “Confidential to Editor” section, and submit your "Accept" recommendation.

Reviewer #1: All comments have been addressed

Reviewer #2: All comments have been addressed

2. Is the manuscript technically sound, and do the data support the conclusions?

Reviewer #1: Yes

Reviewer #2: Yes

3. Has the statistical analysis been performed appropriately and rigorously? 

Reviewer #1: Yes

Reviewer #2: Yes

4. Have the authors made all data underlying the findings in their manuscript fully available?

Reviewer #1: Yes

Reviewer #2: Yes

5. Is the manuscript presented in an intelligible fashion and written in standard English?

Reviewer #1: Yes

Reviewer #2: Yes

6. Review Comments to the Author

Reviewer #1: The authors have responded to all of my concerns in a satisfactory manner. I have no additional concerns.

Reviewer #2: This current version is appropriate to be published. The second revision improved the quality of manuscript and contemplated an extensive analysis of data regarding to circulating galectins and biomarkers of IBD. Possible correlation between galectins and inflammatory bowel diseases (IBD), such as Crohn's disease and ulcerative colitis, is extremely relevant. Moreover, the potential to use galectin levels as biomarkers was clearly demonstrated in this manuscript. Given that authors satisfactorily responded the reviewer's comments, current format is recommended to publication.

7. PLOS authors have the option to publish the peer review history of their article (what does this mean?). If published, this will include your full peer review and any attached files.

Reviewer #1: No

Reviewer #2: Yes: Felipe Leite de Oliveira

---

## [Author Response · Author response to Decision Letter 2]

9 Dec 2019

Dear Dr. Roger Chammas,

Thanks for sending us the reviewers’ comments on the second round revision of this manuscript. We are pleased to see that both reviewers are happy with the second revision and have recommended publication of the manuscript, see below:

Reviewer #1: The authors have responded to all of my concerns in a satisfactory manner. I have no additional concerns.

Reviewer #2: This current version is appropriate to be published. The second revision improved the quality of manuscript and contemplated an extensive analysis of data regarding to circulating galectins and biomarkers of IBD. Possible correlation between galectins and inflammatory bowel diseases (IBD), such as Crohn's disease and ulcerative colitis, is extremely relevant. Moreover, the potential to use galectin levels as biomarkers was clearly demonstrated in this manuscript. Given that authors satisfactorily responded the reviewer's comments, current format is recommended to publication.

The point about CRP and galectin-1/-3 mentioned in your decision letter was raised by Reviewer 1 in the first round review which had been adequately addressed in the second revision as the reviewer clearly stated in their comments. Both reviewers also made it clear that the second revision is satisfactory and no further revision is required. 

It looks that the reviewers’ comments in the first review round somehow corrupted into the system of the second review round which created the confusion for the request of a point already satisfactorily addressed in the new decision letter. 

We hope this letter helps to clarify the situation and the manuscript is ready to be accepted for publication in PLOS One. 

Yours sincerely,

Sreedhar Subramanian, MD, MRCP

---

## [Editor Report · Decision Letter 3]

17 Dec 2019

SERUM GALECTINS AS POTENTIAL BIOMARKERS OF INFLAMMATORY BOWEL DISEASES

PONE-D-19-23084R3

Dear Dr. Subramanian,

We are pleased to inform you that your manuscript has been judged scientifically suitable for publication and will be formally accepted for publication once it complies with all outstanding technical requirements.

With kind regards,

Roger Chammas, M.D, Ph.D

Academic Editor

PLOS ONE
---

## [Editor Report · Acceptance letter]

31 Dec 2019

PONE-D-19-23084R3 

Serum galectins as potential biomarkers of inflammatory bowel diseases 

Dear Dr. Subramanian:

I am pleased to inform you that your manuscript has been deemed suitable for publication in PLOS ONE. Congratulations! Your manuscript is now with our production department. 

With kind regards,

on behalf of

Prof. Roger Chammas 

Academic Editor

PLOS ONE